# Effect of Elevated CO_2_ on Biomolecules’ Accumulation in Caraway (*Carum carvi* L.) Plants at Different Developmental Stages

**DOI:** 10.3390/plants10112434

**Published:** 2021-11-11

**Authors:** Hamada AbdElgawad, Mohammad K. Okla, Saud S. Al-amri, Abdulrahman AL-Hashimi, Wahida H. AL-Qahtani, Salem Mesfir Al-Qahtani, Zahid Khorshid Abbas, Nadi Awad Al-Harbi, Ayman Abd Algafar, Mohammed S. Almuhayawi, Samy Selim, Mohamed Abdel-Mawgoud

**Affiliations:** 1Botany and Microbiology Department, Faculty of Science, Beni-Suef University, Beni-Suef 62521, Egypt; 2Laboratory for Molecular Plant Physiology and Biotechnology, Department of Biology, University of Antwerp, Groenenborgerlaan 171, 2020 Antwerp, Belgium; 3Botany and Microbiology Department, College of Science, King Saud University, Riyadh 11451, Saudi Arabia; saualamri@ksu.edu.sa (S.S.A.-a.); aalhashimi@ksu.edu.sa (A.A.-H.); aghafr@ksu.edu.sa (A.A.A.); 4Department of Food Sciences & Nutrition, College of Food & Agriculture Sciences, King Saud University, Riyadh 11451, Saudi Arabia; wahida@ksu.edu.sa; 5Biology Department, University College of Taymma, Tabuk University, Tabuk 71491, Saudi Arabia; salghtani@ut.edu.sa (S.M.A.-Q.); or Zahid104@yahoo.com (Z.K.A.); nalharbi@ut.edu.sa (N.A.A.-H.); 6Biology Department, College of Science, Tabuk University, Tabuk 71491, Saudi Arabia; 7Department of Medical Microbiology and Parasitology, Faculty of Medicine, King Abdulaziz University, Jeddah 21589, Saudi Arabia; msalmuhayawi@kau.edu.sa; 8Department of Clinical Laboratory Sciences, College of Applied Medical Sciences, Jouf University, Sakaka 72388, Saudi Arabia; sabdulsalam@ju.edu.sa; 9Department of Medicinal and Aromatic Plants, Desert Research Centre, Cairo 11753, Egypt; Mohamed_drc@yahoo.com

**Keywords:** high CO_2_, caraway plants, sprouting, mature plants, nutritious metabolites, antioxidant, antimicrobial activity

## Abstract

Caraway plants have been known as a rich source of phytochemicals, such as flavonoids, monoterpenoid glucosides and alkaloids. In this regard, the application of elevated CO_2_ (eCO_2_) as a bio-enhancer for increasing plant growth and phytochemical content has been the focus of many studies; however, the interaction between eCO_2_ and plants at different developmental stages has not been extensively explored. Thus, the present study aimed at investigating the changes in growth, photosynthesis and phytochemicals of caraway plants at two developmental stages (sprouts and mature tissues) under control and increased CO_2_ conditions (ambient CO_2_ (a CO_2_, 400 ± 27 μmol CO_2_ mol^−1^ air) and eCO_2_, 620 ± 42 μmol CO_2_ mol^−1^ air ppm). Moreover, we evaluated the impact of eCO_2_-induced changes in plant metabolites on the antioxidant and antibacterial activities of caraway sprouts and mature plants. CO_2_ enrichment increased photosynthesis and biomass accumulation of both caraway stages. Regarding their phytochemical contents, caraway plants interacted differently with eCO_2_, depending on their developmental stages. High levels of CO_2_ enhanced the production of total nutrients, i.e., carbohydrates, proteins, fats and crude fibers, as well as organic and amino acids, in an equal pattern in both caraway sprouts and mature plants. Interestingly, the eCO_2_-induced effect on minerals, vitamins and phenolics was more pronounced in caraway sprouts than the mature tissues. Furthermore, the antioxidant and antibacterial activities of caraway plants were enhanced under eCO_2_ treatment, particularly at the mature stage. Overall, eCO_2_ provoked changes in the phytochemical contents of caraway plants, particularly at the sprouting stage and, hence, improved their nutritive and health-promoting properties.

## 1. Introduction

Sprouts are the young seedlings produced through seed germination; they have become more popular vegetables and functional foods in most western countries [1]. Sprouts are rich in active phenolic compounds, proteins, minerals, vitamins and glucosinolates. Thus, they are considered as healthy foods with higher antioxidant activities [2,3].

Caraway (*Carum carvi* L.), has been known as an important medicinal plant, being cultivated worldwide. Caraway plants have been used as a remedy for various ailments, such as diarrhea, asthma, cholera and hypertension [4]. In addition, caraway fruits have been prescribed in herbal mixtures as a drug for digestive, carminative and lactogenic disorders. Moreover, caraway have been used as antiallergic, antibacterial, anthelmintic [5], antifungal [6], bronchodilator [7] and cholinergic [8] agents. Such versatile biological activities are mainly due to the existence of multiple phyto-constituents including steroptin, cumene, thymene, amino acids such as lysine and threonine, calcium, iron, starch, tannins and dietary fibers [9]. The essential oils of caraway seeds contain thymol, γ-terpinene, *p*-cymene, β-pinene, α-pinene and carvacrol, in addition to monoterpenes such as carvone and limonene, which usually make up 95% of the caraway essential oils [10,11]. Thus, caraway essential oils have largely been used as a raw material in pharmaceutical industries [12].

Thus, in view of the wide traditional uses of caraway plants, several strategies have been approved to improve the growth, bioactivity and nutritive values of *Carum* sprouts [13,14]. For example, different innovative approaches such as laser light, UV-B irradiation [15,16] and high atmospheric CO_2_ were recommended to enhance plant growth and nutritive values [17,18]. Previous studies have reported that increased atmospheric CO_2_ levels significantly influenced the plant chemical composition by changing carbon and nitrogen metabolism [19,20], which in turn increases the availability of phytochemicals required for growth [21]. Moreover, eCO_2_ was demonstrated to enhance photosynthesis, leading to biosynthesis of sugars, which could be then broken down in the dark respiration process [20,22]. Subsequently, these processes are assumed to provide precursors and energy needed for the biosynthesis of active metabolites. Moreover, eCO_2_ was found to enhance the biological activities of many plant species, such as peppermint, basil, dill and parsley, as well as lemongrass, brassica and alfalfa sprouts [14,17,23,24,25]. To the best of our knowledge, the influence of eCO_2_ on caraway plants and associated metabolic changes have not been previously investigated. Thus, this study evaluates the effect of eCO_2_ treatment on the nutritive values, phytochemicals (i.e., pigments, phenolics, flavonoids, vitamins, mineral profiles and essential oil yields) and antioxidant activities of both caraway sprouts and mature plants. Therefore, this study was performed, not only to enhance the sprouting conditions under eCO_2_ but also to obtain insights into how eCO_2_ improves the sprouting process, and to compare the significant differences between caraway sprouts and mature plants regarding their response to the effect of eCO_2_ on their phytochemical contents and biological activities.

## 2. Results

### 2.1. Growth and Biomass Production of Caraway Sprouts and Mature Plants as Affected by eCO_2_

The present investigation has clearly shown that eCO_2_ treatment has exerted significant increases in biomass (fresh and dry weights) in the examined sprouts and mature plants when compared to controls (Figure 1). Meanwhile, the effect of eCO_2_ seemed to be more pronounced on caraway mature plants than the sprouts. On the other hand, eCO_2_ significantly enhanced the accumulation of photosynthetic pigments in caraway sprouts and mature plants, whereas significant increments in chlorophyll a, b and (a + b) were obtained for both sprouts and mature plants (Figure 1).

### 2.2. Impact of eCO_2_ on Minerals and Primary Metabolites of Caraway Sprouts and Mature Plants

In the current study, the impact of eCO_2_ on total nutrients, i.e., primary (lipid, proteins, sugars and crude fibers) and secondary metabolites (flavonoids, phenols and steroids) was investigated in both caraway sprouts and mature plants. The treatment with eCO_2_ significantly increased most of the detected total nutrients in sprouts and mature plants, as compared to controls (Figure 2). Moreover, we determined the levels of mineral elements, i.e., macrominerals (Ca, K, P, Mg, N and Na) and microminerals (Cu, Fe and Zn) in both plant stages (Figure 2). Predominantly, higher levels of CO_2_ increased most of the detected elements in sprouts and mature plants in comparison to controls, except for Na, N and Zn. The predominant macrominerals observed in both stages were K and P, while the lowest amount was reported for Zn.

Regarding their organic acids content, the investigated sprouts and mature plants contained comparable levels of six detected organic acids under control conditions, i.e., malic, succinic and citric acids (predominant), as well as oxalic, isobutyric and fumaric acids (Table 1). eCO_2_ treatment showed significant increases in most of the organic acids detected in both stages.

The current results also revealed that both untreated caraway sprouts and mature plants had almost the same amino acids profile, whereas glutamine and glutamic acid were detected as the major amino acids in both caraway stages (Table 1). Treatment with eCO_2_ significantly increased the concentrations of some detected amino acids in sprouts and mature plants, while others were not affected, when compared to controls.

### 2.3. Secondary Metabolites of Caraway Sprouts and Mature Plants as Influenced by eCO_2_

In addition, several essential oils were estimated in control and eCO_2_-treated caraway plants, whereby thyme, γ-terpinene, *p*-cymene, carvacrol and β-pinene were the most prominent essential oils, followed by lower amounts of other components (Figure 2). The results showed that eCO_2_ significantly decreased the levels of essential oils in both sprouts and mature plants compared to normally cultivated controls. On the other hand, a considerable amount of several vitamins were determined in caraway sprouts and mature plants (Figure 2). eCO_2_ treatment significantly increased almost all investigated vitamins in both caraway stages.

The phenolic profile of caraway sprouts and mature plants was almost similar, whereas gallic acid and quercetin were quantified as the most dominant phenolic acids and flavonoids, respectively, in both stages (Figure 3). eCO_2_ caused significant increments in the phenolic and flavonoid profiles of sprouts when compared to controls.

### 2.4. Effect of eCO_2_ on Antioxidant Activities of Caraway Sprouts and Mature Plants

Antioxidant activities of caraway sprouts and mature plants were estimated in our study (by ferric reducing antioxidant power (FRAP), DPPH• radical, and lipid peroxidation assays), following eCO_2_ treatment (Figure 3). The results showed that eCO_2_ had a more enhancing effect on the antioxidant activities of sprouts than the mature plants, when compared to their respective controls.

### 2.5. Antimicrobial Activity of Caraway Sprouts and Mature Plants under eCO_2_ Treatment

In the current investigation, the effect of eCO_2_ on the antimicrobial activity of caraway sprouts and mature plants was determined. The test was conducted against a set of 13 microbes, based on the minimal inhibitory concentration (MIC) of the selected caraway extracts (Table 2). As indicated by the diameter of inhibition zone, the results showed that treatment with eCO_2_ significantly increased the antimicrobial potency of caraway sprouts and mature plants predominantly against *Staphylococcus epidermidis*, *Staphylococcus saprophyticus*, *Streptococcus salivarius*, *Enterococcus faecalis*, *Salmonella typhimurium*, *Pseudomonas aeruginosa*, *Candida glabrata*, *Serratia marcescens* and *Escherichia coli* (Table 2). Moreover, eCO_2_ significantly increased the antimicrobial activity of mature caraway plants against *Proteus vulgaris*. The increased antibacterial activity might be attributed to enhanced levels of phytochemicals in eCO_2_-treated caraway sprouts and/or mature tissues. The increased levels of photosynthetic pigments might also be involved in antimicrobial effects against some bacteria such as *Vibrio parahaemolyticus*, *Micrococcus luteus*, *Bacillus subtilis* and *Escherichia coli*. However, under eCO_2_ treatment, there were significant reductions in the antimicrobial activity of caraway sprouts and mature plants against *Enterobacter aerogenes*. In addition, eCO_2_ significantly reduced the antimicrobial activity of caraway sprouts against *Proteus vulgaris* and *Candida albicans*, compared to normally cultivated controls. Meanwhile, eCO_2_ had no impact on the antimicrobial activity of both stages against *Aspergillus flavus*, and, also, no change was observed in the antimicrobial activity of eCO_2_-treated mature plants against *Candida albicans*. It can be noted that the effect of eCO_2_ was more pronounced on the antimicrobial activities of sprouts than their mature plants.

## 3. Discussion

Our study has demonstrated the importance of eCO_2_ to enhance the nutritive and health-promoting values of plants, particularly in the context of rising atmospheric CO_2_. Depending on their developmental stages, plants might interact differently in response to eCO_2_. Preferably, sprouts can be good candidates for the positive impact of eCO_2_ on their nutritious metabolites. This can be supported by our previous studies and other research that also discussed the application of eCO_2_ to increase the food functional values of many plants and sprouts [2,3,14,16,24]. Thus, the expected availability of eCO_2_ in the future could be a promising approach to stimulate the growth and phytochemicals of sprouts.

### 3.1. eCO_2_ Enhanced Biomass Production of Caraway Sprouts and Mature Plants

It has been known that eCO_2_ could effectively enhance photosynthesis and plant growth [23]. In our study, the eCO_2_-induced biomass accumulation in both sprouts and mature plants could be the result of enhanced photosynthetic activity under eCO_2_ treatment, which triggered the biosynthesis of sugars and organic acids, and finally resulted in enhanced growth and biomass accumulation [20,23]. Similar to our results, eCO_2_ has reportedly increased photosynthesis and the growth of many plants and sprouts, e.g., tobacco, tomato, peppermint, *Arabidopsis*, *basil*, *Hymenocallis littoralis*, *Isatis indigotica*, broccoli and alfalfa sprouts [20,23,24,25,26,27,28]. In addition, eCO_2_ enhanced the yield of some crops, such as carrot, radish and turnip [22]. Moreover, CO_2_ enrichment has been found to increase the levels of sugars and biomass production in dill and parsley [17]. Furthermore, the use of other elicitors, (e.g., laser light) to stimulate the growth and biomass accumulation of buckwheat sprouts has been recently explored [16].

On the other hand, eCO_2_ was observed to enhance the photosynthetic pigment content in caraway sprouts and mature plants (Figure 1). It can be noted that caraway mature plants are likely to better respond to eCO_2_, thus accumulating higher photosynthetic pigments than the sprouts. Similar to our results, eCO_2_ has been recently applied to enhance chlorophyll content (a, b) in alfalfa sprouts [25]. In this regard, the eCO_2_-induced chlorophyll content is assumed to enhance much more photosynthesis, which, in turn, leads to more sugar accumulation and, hence, higher biomass production [20].

### 3.2. eCO_2_ Similarly Affected the Primary Chemical Composition of Caraway Sprouts and Mature Tissue

High CO_2_ levels could modify the plant chemical composition by changing the carbon and nitrogen metabolism [20]. Herein, the treatment of caraway plants with eCO_2_ significantly increased most of the primary metabolites detected in sprouts and mature plants, as compared to controls (Figure 2). Both of the plant stages seemed to equally respond to the effect of eCO_2_ on their nutrients. The results are in agreement with those obtained by [23] who reported a significant increase in total lipids, carbohydrates and proteins in peppermint and basil following treatment with eCO_2_. Additionally, [29] reported that eCO_2_ caused significant increments in the sugars, proteins and phenolic compounds of oregano. Several studies have demonstrated the enhancing effects of high atmospheric CO_2_ on increasing the nutritive values, sugar and phytochemical contents of many plants’ tissues and sprouts, e.g., dill and parsley, as well as the total proteins, lipids, carbohydrates and fibers of alfalfa sprouts [17,25]. In this regard, the accumulation of sugars, in response to elevated photosynthesis, is expected to provide the energy necessary for the biosynthesis of multiple classes of metabolites [20,26,28].

At individual levels, the present study has shown eCO_2_ to increase most of the amino acids and organic acids detected in both caraway stages, whereas the sprouts and mature plants showed a similar response to eCO_2_. In accordance with this, it has been observed that some organic acids were accumulated, while others were not affected, in peppermint, basil, parsley and dill treated with eCO_2_ [17,23]. Regarding the amino acid contents, both caraway sprouts and mature plants responded in a similar manner to eCO_2_ effects on their amino acids (Table 1). Similar to our results, eCO_2_ has been shown to enhance almost all amino acids in lemongrass sprouts, broccoli sprouts, dill, basil, parsley and peppermint [14,17,23,24]. Meanwhile, eCO_2_ was incapable of increasing the biosynthesis of L-canavanine; a non-protein amino acid in alfalfa sprouts [25]. Such increases in amino acids could be due to the ability of eCO_2_ to induce the availability of C skeleton and metabolic energy needed for the biosynthesis of amino acids [28,30]. In addition, the variation in amino acid contents in sprouts might be related to the higher rates of physiological activities during the sprouting process. Some amino acids, reported herein, have been investigated for their health effects, e.g., glutamine has been involved in fighting some neurodegenerative diseases such as Alzheimer’s diseases, besides being an essential factor for lymphocyte proliferation and the production of cytokines [31].

### 3.3. eCO_2_ Differentially Affected the Mineral and Secondary Chemical Compostion of Caraway Sprouts and Mature Tissue

Sprouts have been considered as rich sources of vitamins, phenolics and minerals, whose deficiency could risk human health [32]. Therefore, increasing their contents in caraway sprouts and mature plants by using promising approaches, such as eCO_2_, might enhance their nutritional and health-promoting values. In this study, the effect of eCO_2_ on mineral content was more pronounced in sprouts than mature plants (Figure 2). Matched with our results, alfalfa sprouts have been observed to accumulate more macronutrients (Na, Mg, K, P and Ca) and micronutrients (Fe, Zn, Cu and Mn), when treated with eCO_2_ [25]. In addition, significant increments in the levels of Ca, Mg, Zn, K, Cu and Cd were detected following eCO_2_ treatment of many plant species such as basil and peppermint [17,23]. Moreover, it has been reported that Ca, Mn and Cu were significantly improved in carrot and radish when treated with eCO_2_ [22]. Previous studies also reported enhancing effects of eCO_2_ on primary metabolism, nutrient uptake and the levels of minerals in many plant species [28,33,34]. However, eCO_2_ did not induce significant effects on most minerals previously detected in dill and parsley [17], besides its negative impact on minerals in many plants [35,36,37]. Such influence might be ascribed to the dilution effect rather than decreased nutrient absorption [17].

At the antioxidant level, similar responses to eCO_2_ were recorded for phenolics and flavonoids, where caraway sprouts seemed to respond better to the eCO_2_ effects on phenolic content than the mature plants (Figure 3). Caraway seeds, fruits and sprouts, have been known to contain a diverse amount of flavonoids, monoterpenoid glucosides, lignins, alkaloids and polyacetylenic compounds [38,39,40,41,42,43]. In the current study, eCO_2_ caused significant increments in the phenolic and flavonoid profiles of caraway sprouts. In agreement, eCO_2_ has been reported to enhance the phenolic content of alfalfa sprouts, basil, peppermint, parsley, dill and fenugreek [17,23,25,29]. The improved levels of phenols and flavonoids in eCO_2_-treated caraway sprouts could be attributed to the abundance of C and N intermediates that could be used for the biosynthesis of these phytochemicals, given that eCO_2_ has been reported to affect the metabolism of C and N [28].

The impact of eCO_2_ on vitamins and essential oil levels in plants is poorly studied, whereas limited studies have reported improved vitamin C and E contents in eCO_2_-treated medicinal plants [23]. Similarly, positive effects of eCO_2_ on some vitamins, e.g., vitamin K1, A, B, C and E, were recorded in basil, peppermint, parsley, orange, strawberry and dill as well as alfalfa sprouts [17,23,25,44]. Higher contents of essential oils were reportedly present in caraway, particularly in seeds, whereas the most prominent essential components were carvone and limonene, beside trace amounts of acetaldehyde, camphene, furfural, pinene and phellandrene [45,46,47,48]. In our study, the change in essential oil content following treatment with high eCO_2_ might be affected by some factors such as the characteristics and tillage of soil, annual precipitation, fertilization amount, breeding, maturation and harvesting [49,50]. It was also previously reported that the composition and the amount of essential oils and vitamins depend on climatic conditions during fruit formation and ripening [51].

### 3.4. eCO_2_ Treatments Induced Higher Biological Activities in Sprouts than in Mature Tissues of Caraway Plants

Caraway plants have been recommended in medicinal uses as antioxidant, antimicrobial, immune-modulatory, antiulcerogenic and antispasmodic as well as antioxidative agents. Such various biological activities could be due to their richness in polyphenolic compounds, flavonoids, essential oils, organic and amino acids, and vitamins [52,53,54,55,56]. Herein, eCO_2_ was observed to exert an enhancing effect on the antioxidant activities of both sprouts and mature plants. Such results could indicate that mature plants might be more responsive than sprouts to eCO_2_ effects on their antioxidant activities.

In accordance, eCO_2_ has previously been found to increase the antioxidant activities of many plants and sprouts, e.g., alfalfa sprouts, lemongrass sprouts, *Medicago lupulina*, basil, peppermint, parsley, dill, fenugreek, strawberry, *Citrus aurantium* and *Labisia pumila* [14,17,23,25,29,44,57,58,59,60]. Such eCO_2_-induced antioxidant effects could be related to the increased levels of several phenolic compounds accumulated in response to eCO_2_. Additionally, in ginger plants treated with eCO_2_, a significantly similar improvement in the levels of phenolic compounds and vitamin C was detected, which increased the antioxidant capacity [44]. Previous studies reported that there was a significant link between antioxidant activity and polyphenol and flavonoids contents, as well as vitamins such as α-tocopherol, which collectively increases the radical scavenging activity of the essential oils of plants and vegetables [61,62,63,64]. Recently, caraway essential oils have shown a high oxidative stability and antioxidant properties [65]. This could be due to their high contents of unsaturated fatty acids, sterols, α and γ-tocopherols, and phenolic compounds. The improved phenolics, flavonoids and other constituents significantly improved the antioxidant and anti-lipid peroxidation in our studied plants in a similar behavior to other previously reported studies on fenugreek [29]. Moreover, the antioxidant capacity of some minerals detected herein, e.g., Cu, Mn, Zn and Se might be improved following treatment of caraway plants with eCO_2_ [66].

In the current investigation, the effect of eCO_2_ was more pronounced on the antimicrobial activities of sprouts than their mature plants. Supporting our results, previous studies reported that caraway essential oils play a potential role in the growth inhibition of many bacteria and fungi such as *Vibrio cholera*, *Escherichia coli*, *Staphylococcus aureus*, *Salmonella typhi* and *Mycobacterium tuberculosis* [67,68]. Similar antimicrobial effects were determined for caraway essential oils, particularly carvone, against the growth of bacteria and fungi during potato storage, such as *Fusarium sulphureum* and *Phoma exigua* [69,70,71]. eCO_2_ has also been shown to enhance the antibacterial effects of fenugreek against *Staphylococcus aureus*, *Bacillus subtilis* and *Streptococcus* sp. [29]. Moreover, the antifungal potency of caraway oil was previously reported against *Alternaria alternata*, *Fulvia fulvium, Cladosporium cladosporioides*, and *Phoma macdonaldii* [72].

The observed antimicrobial activities can be explained by higher quantities of polyphenolic compounds along with appreciable amounts of vitamins such as α-tocopherol [61,62,63]. It could also be suggested that eCO_2_ increased the antimicrobial activity of eCO_2_-treated caraway sprouts and mature plants, which might be related to the improved content of phytochemicals such as phenolic acids, flavonoids, vitamins and mineral contents, which consequently enhance the antibacterial activities, as previously reported for fenugreek [29]. The enhanced levels of photosynthetic pigments might also be involved in antimicrobial effects against *Vibrio parahaemolyticus*, *Micrococcus luteus*, *Bacillus subtilis* and *Escherichia coli*. Similarly, it has been previously demonstrated that high levels of CO_2_ increased the antibacterial activities of dill, parsley, peppermint and basil against *Streptococcus* spp. and *Escherichia coli*, the activities that could be supported by their phenolic contents [17,23]. Overall, improving the potential antimicrobial activity of caraway oils and extracts could further support their applications in medicinal purposes, food preservation and the cosmetic industry.

### 3.5. Principal Component Analysis (PCA) Confirmed the Developmental Stage-Specific Response of Caraway Plants to eCO_2_

To better understand the developmental stage responses, we performed PCA of the chemical composition and biological activities of sprouts and mature tissues under control and eCO_2_ treatments (Figure 4). We observed a clear separation between the treatment parameters along the PC1, which explains 66% of the total variation. Remarkably, eCO_2_ induced the accumulation of minerals and several components as well as antimicrobial and antioxidant activities. There was also a clear separation between the parameters of sprouts and mature stages along PC2 (represents 18% of the total variation). Caraway plants at the mature stage showed high levels of a few organic and amino acids, phenolic acids including methionine, leucine, velutin, valine, isobutyric and aspartic acids and the flavonoid luteolin, as previously discussed [9,52,53,54,55,56]. However, caraway sprouts were richer than mature plants, in many of the essential oils, phenolics, ash, mineral (Zn, N), and amino acids (proline and glycine) as well as antioxidant (DPPH•) and antibacterial activity against *P. aeruginosa*, *S. marcescens*, *P. vulgaris* and *E*, *aerogenes*. Together, these data showed that caraway plants at different developmental stages were differentially grouped, indicating the specificity of nutritive metabolite accumulation in response to eCO_2_ treatment.

## 4. Materials and Methods

### 4.1. Plant Material and Growth Conditions

Seeds of caraway (*Carium carvum* L.) were collected from Agricultural Research Center (Giza, Egypt). The seeds were washed with distilled water and soaked for 1 h in 5 g L^−1^ sodium hypochlorite, then they were kept overnight in distilled water. For sprouting process, the seeds were distributed on trays filled with vermiculite and irrigated with Milli-Q water every two days. For growing plants till mature stage, the seeds were sown in loamy soil and organic compost (50:50%) in pots and the soil water content (SWC) was adjusted to 60%. The growth conditions were adjusted to 25 °C air temperature, a 16/8-h day/night photoperiod using white fluorescent tubes with photosynthetically active radiation (400 µmol m^−2^ s^−1^ and 60% humidity). According to IPCC-SRES B2-scenario prediction of elevated CO_2_ of the year 2100, the seeds were maintained under two climate conditions, (1) ambient CO_2_ (a CO_2_, 400 ± 27 μmol CO_2_ mol^−1^ air); (2) elevated CO_2_ (eCO_2_, 620 ± 42 μmol CO_2_ mol^−1^ air ppm). The sprouts and mature tissues from each treatment were harvested after 9 and 45 days and weighed, then they were frozen in liquid nitrogen and kept at −80 °C for biochemical analyses. Each experiment was replicated at least two times, and for all assays, 3 to 5 replicates were used and each replicate corresponded to a group of sprouts and mature plants harvested from a certain tray.

### 4.2. Determination of Photosynthetic Rate

Photosynthesis (μmol CO_2_ m^−2^ s^−1^) was detected by EGM-4 infrared gas analyzer (PP Systems, Hitchin, UK). Photosynthetic rate was detected from 180 s measurements of net CO_2_ exchange (NE).

### 4.3. Pigment Analysis

Caraway samples were homogenized for 1 min at 7000 rpm in acetone by using a MagNA Lyser (Roche, Vilvoorde, Belgium), then centrifugation was performed at 4 °C, 14,000× *g* for 20 min. The supernatant was separated then filtered (Acrodisc GHP filter, 0.45 μm 13 mm). Thereafter, analysis of the obtained solution was conducted by using HPLC (Shimadzu SIL10-ADvp, reversed-phase, at 4 °C) [16]. Extraction of chlorophyll a and b was performed, then quantified by using a diode array detector (Shimadzu SPDM10Avp) at four wavelengths (420, 440, 462 and 660 nm).

### 4.4. Preparation of Caraway Extracts

An ETA 0067 grinder with grinding stones, a VIPO grinder and a Vibrom S2 (Jebavý, Trebechovice p. O., CR) cryogenic grinder (liquid nitrogen) were tested for sample homogenization. An SE-1 (SeKo-K, Brno, CR) extractor for supercritical fluid extraction (SFE) and an apparatus for steam distillation according to CSN 58-0110 and CSN 6571 were applied for extraction and subsequent determination of the total content of caraway oils. Approximately 500 mg of exactly weighed fresh sample was transferred into an extraction column for SFE extraction. The extraction was performed for 60 min at 40 MPa, extractor temperature 80 °C and restrictor temperature 120 °C. The extract was trapped into a hexane layer in a trapping vessel.

### 4.5. Analysis of Mineral Contents

Macro and microelements were evaluated according to [18]. About 200 mg from both eCO_2_-treated and non-treated caraway sprouts and mature plants were digested in HNO_3_/H_2_O solution (5:1 *v*/*v*) in an oven. Then, the concentrations of macrominerals and trace elements were estimated at 25 °C by using inductively coupled plasma mass spectrometry (ICP-MS, Finnigan Element XR, and Scientific, Bremen, Germany), where nitric acid (1%) was used as a standard.

### 4.6. Measurement of Phenolic Acids, Flavonoids and Vitamins

Chromatographic techniques such as HPLC were used to measure the levels of phenolic acids, flavonoids and vitamins in eCO_2_-treated and non-treated caraway sprouts and mature plants as previously described in [16,73]. Identification of compounds was conducted by comparing the standard mixtures to the relative retention time of each compound from each sample. The peak area of the corresponding standard was used to calculate the concentration of each compound. Phenolic compounds were measured by HPLC. A total of 50 mg of frozen dried samples were mixed with 4:1 *v*/*v* acetone–water solution. HPLC system (SCL-10 AVP, Japan), supplied with a LiChrosorb Si-60, 7 μm, 3 × 150 mm column, diode array detector) was used for determination of flavonoids and phenolic acids, whereas the mobile phase consisted of (90:10) water/formic acid, and (85:10:5) acetonitrile/water/formic acid, at 0.8 mL/min (flow rate), while 3,5-dichloro-4-hydroxybenzoic was used as an internal standard. The concentration of each detected compound was evaluated by using the peak area of the corresponding standard. The contents of thiamine and riboflavin were determined in caraway samples by using UV and/or fluorescence detectors [16]. Separation was conducted on a reverse-phase (C18) column (HPLC, methanol–water).

### 4.7. Measurement of Antioxidant Capacity

The determination of antioxidant capacity was conducted through ferric reducing antioxidant power (FRAP), DPPH• radical, and lipid peroxidation assays as previously reported [14,74,75,76].

#### 4.7.1. Ferric Reducing Antioxidant Power (FRAP) Method

About 0.2 g of caraway sample was extracted in ethanol (80%), and then centrifugation was performed (14,000 rpm, 20 min). Afterwards, 0.1 mL of extracts was added to 0.25 mL of FRAP reagent (20 mM FeCl_3_ in 0.25 M acetate buffer, pH 3.6) at room temperature, and, finally, the absorbance was detected at 517 nm [14].

#### 4.7.2. DPPH• Assay

DPPH· radical has a violet color with a maximum absorbance at 517 nm that becomes colorless in presence of antioxidants. The reaction mixtures consisted of 0.1 mL of samples and 3.9 mL of 200 μM DPPH• (prepared in ethanol), and incubated in dark (30 min, 37 °C). Afterwards, the absorbance was detected (517 nm). The percentage of inhibition was calculated against a control [74].

#### 4.7.3. Lipid Peroxidation Assay

Lipid peroxidation was determined through detection of MDA present in caraway samples of eCO_2_-treated and control plants. In this assay, a colorimetric lipid peroxidation (MDA) assay kit (MAK085, 3050 Spruce Street, Saint Louis, MO 63103, USA) was used, whereas the MDA in the sample reacts with thiobarbituric acid (TBA) to generate a MDA-TBA adduct, which can be easily quantified calorimetrically (OD = 532 nm) or fluorometrically (Ex/Em = 532/553 nm). This assay detects MDA levels as low as 1 nmol/well calorimetrically and 0.1 nmol/well fluorometrically [14].

### 4.8. Determination of Total Carbohydrates, Protein, Lipids and Fibers

Nelson’s method was used to measure carbohydrates from each caraway sample (eCO_2_-treated and control plants). Concentration of protein was detected for each frozen caraway sample (0.2 g FW) according to Lowry methods [75]. Detection of total lipids was conducted based on Folch method modified by [76], whereas the samples were homogenized in chloroform/methanol (2:1). Afterwards, centrifugation was performed at 3000× *g* for 15 min. A rotary evaporator was used to evaporate the chloroform phase containing lipids., and then the pellets were redissolved in a mixture of toluene/ethanol (4/1 *v*/*v*). A saline solution was mixed with the extract. The extracted lipids were concentrated by a rotary evaporator and then weighed in vials to calculate the total lipid content. Fibers also were extracted from the target samples and evaluated according to AOAC (1990), where α-amylase was used for sample gelatinization (30 min, pH 6, 100 °C), then protease was used for enzymatic digestion (30 min, pH 7.5, 60 °C). Thereafter, amyloglucosidase was used for proteins and starch removal (30 min, pH 6 and 0 °C). Finally, fibers were precipitated with ethanol, and the residue was weighed after washing.

### 4.9. Analysis of Essential Oils, Organic Acids, and Amino Acids

#### 4.9.1. Analysis of Essential Oils

The steam distillation method, according to the standards CSN 58 0110 and CSN 6571, was used. Depending on the expected content of essential oil, an exactly weighed sample (10–25 g) was transferred into a distillation vessel, and then 400 mL of water and boiling stones were added. The samples were boiled for 4 h. Then, cooling was stopped, and distillation was prolonged for a while until all essential oils were quantitatively transferred into a calibrated tube. Then, the heating was stopped and the volume of the extracted essential oils was measured after 5 min. The extracted or distilled samples were stored in a refrigerator at 1–4 °C (for 2 days), if necessary, and analyzed by GC. A gas chromatograph HP 4890D (Hewlett Packard) with a FID detector was used for determination of limonene-to-carvone ratio in the samples. Separation was performed using an HP-5 (Crosslinked 5% PH ME Siloxane, 15 m × 0.53 mm × 1.5 μm film) column at helium flow rate 2 mL/min, injector temperature 220 °C and detector temperature 240 °C using temperature program 60 °C, 40 °C/min up to 220 °C, 2 min at 220 °C. Portions of 2 μL of each essential oil (dissolved in hexane) were injected into the used analytical column. Resulting chromatograms were treated using CSW (Data Apex, Prague, CR) data station [77]. Identification of oil components was achieved based on their retention indices (RI, determined with reference to a homologous series of normal alkanes) and by comparison of their mass spectral fragmentation patterns (NIST) database (G1036A, revision D.01.00)/Chem-Station data system (G1701CA, version C.00.01.08)].

#### 4.9.2. Organic Acids Analysis

Organic acids were detected in caraway extracts by using HPLC, isocratically, with 0.001 N sulfuric acid, at 210 nm and flow of 0.6 mL min^−1^ [78]. The assay was performed by using liquid chromatographer (Dionex, Sunnyvale, CA, USA) with LED detector Ultimate 3000. The latter cooperated with the following devices: pump (LPG-3400A) EWPS-3000SI autosampler, TCC-3000SD column thermostat and the Chromeleon v.6.8 computer software. Meanwhile, the separation was conducted by using Aminex HPH-87 H (300 × 7.8 mm) column with IG Cation H (30 × 4.6) pre-column of Bio-Red firm, at a temperature of 65 °C.

#### 4.9.3. Amino Acid Analysis

Detection of amino acid in caraway samples was carried out according to [79]. About 3 mg of each sample was hydrolyzed with 6 M HCl (6 h, 150 °C). Afterwards, the acid was evaporated by rotary evaporation (RE500 Yamato Scientific America Inc.), and the samples were redissolved in 2 mL of sodium citrate buffer (pH 2.2). For sample derivation, phthalaldehyde (OPA) (7.5 mM) was added to samples in citrate buffer (OPA reagent contains β-mercaptoethanol and Brij 35). The HPLC method precision was evaluated using external and internal standards. The amino acid reference standards consisted of 15 amino acids (0.05 µmoles mL^−1^ amino acid), which were used for detection of retention times of each amino acid. Meanwhile, the internal standard (0.05 µmoles mL^−1^ α aminobutyric) was added to the reference sample as well as the plant sample. The gradient mobile phase consisted of 0.1 M sodium acetate and methanol (9:1), while C18 column reversed-phase (100 × 4.6 mm × 1/4″ Microsorb 100-3 C18) was used for sample elution. Fluorescence detection was realized using an excitation–emission wavelength of 360 and 455 nm, respectively. For amino acid peak integration, Star Chromatography (Varian version 5.51) was applied.

### 4.10. Antibacterial Activity of Caraway Extracts

Antibacterial activity of caraway was analyzed by standard dilution in liquid media according to a previously reported methodology [14]. Dimethyl sulfoxide (DMSO) was used to dissolve 100 mg of the essential oil sample. The range of oil dilution from 1 to 20 mg mL^−1^ was prepared in media Mueller-Hinton Broth of Merck. Then, 0.1 mL of 18 h liquid culture of standard strain (*Staphylococcus aureus* ATCC 6538 P) diluted 1:10,000 in the same medium (number of inoculum contained 104–105 bacterial cells in 1 mL), was added to the media. Incubation of the tested samples was conducted in 37 °C for 18 h. The value of MIC (Minimal Inhibitory Concentration) was defined as the lowest concentration of the oil completely inhibiting the growth of standard strain. This value was calculated on antibiotic units (AU), based on that the value of MIC is equivalent to 1AU. The results were referenced to 1 g of oil. The antibacterial activity of the tested samples was evaluated against *Candida albicans* (ATCC90028), *Candida glabrata* (ATCC90030), *Aspergillus flavus* (ATCC9170), *Staphylococcus saprophyticus* (ATCC 19701), *S. epidermidis* (ATCC 12228), *Enterococcus faecalis* (ATCC 10541), *Streptococcus salivarius* (ATCC25975), *E. coli* (ATCC 29998), *Salmonella typhimurium* (ATCC14028), *Pseudomonas aeruginosa* (ATCC10145), *Proteus vulgaris* (ATCC8427), *Enterobacter aerogenes* (ATCC 13048), *Serratia marcescens* (ATCC99006) and *Salmonella typhimurium* (ATCC14028).

### 4.11. Statistical Analyses

Statistical analyses were performed using SPSS statistical package (SPSS Inc., Chicago, IL, USA). Replication of each experiment was carried out (two times). Three to five replicates were used for all assays and each replicate corresponded to a group of sprouts and mature plants harvested from a certain tray. One-way analysis of variance (ANOVA) was performed. Tukey’s test was used as the post-hoc test for the separation of means (*p* < 0.05). Principal component analysis (PCA) was generated by Multi Experimental Viewer (TM4 software package).

## 5. Conclusions

This study reported, for the first time, the application of eCO_2_ to improve the nutritive value, functionality and health-promoting prospective of caraway sprouts and mature plants. The exposure of caraway sprouts and mature plants to high CO_2_ significantly improved the nutritive value, levels of essential minerals, free amino and organic acids, phenolic compounds and other metabolites. In addition, the antimicrobial activity increased; but no change was reported in essential oils’ contents under eCO_2_ treatment. The impact of eCO_2_ was more pronounced on sprouts than mature plants, regarding minerals, vitamins and phenolic compounds, while the mature plants displayed higher antioxidant abilities than that of sprouts, under eCO_2_ treatment. Meanwhile, both caraway sprouts and mature plants responded equally to eCO_2_ effects on their total nutrients, organic and amino acids as well as antimicrobial activities. Thus, under the current scenario of increasing atmospheric CO_2_, eCO_2_ could be applied to improve the nutritive value, functionality and health-promoting prospective of caraway sprouts and mature plants, depending on the developmental stage, and the kind of metabolites and bioactivity needed to be modulated under eCO_2_ treatment.

## Figures and Tables

**Figure 1 plants-10-02434-f001:**
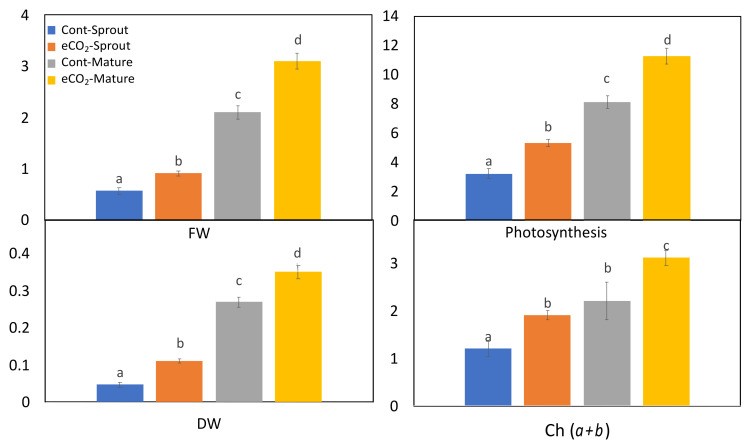
Biomass; fresh weight (FW) (mg/g FW) and dry weight (DW) (mg/g FW), photosynthesis (μmol CO_2_ m^−m^ s^−s^), and pigments content (chlorophyll *a + b*) (mg/g FW) of control and eCO_2_-treated caraway sprouts and mature plants. Data are represented by the means of at least 3 replicates ± standard deviations. Different small letter superscripts (a, b, c and d) within a row indicate significant differences between control and eCO_2_-samples. One-way analysis of variance (ANOVA) was performed. Tukey’s test was used as the post-hoc test for the separation of means (*p* < 0.05).

**Figure 2 plants-10-02434-f002:**
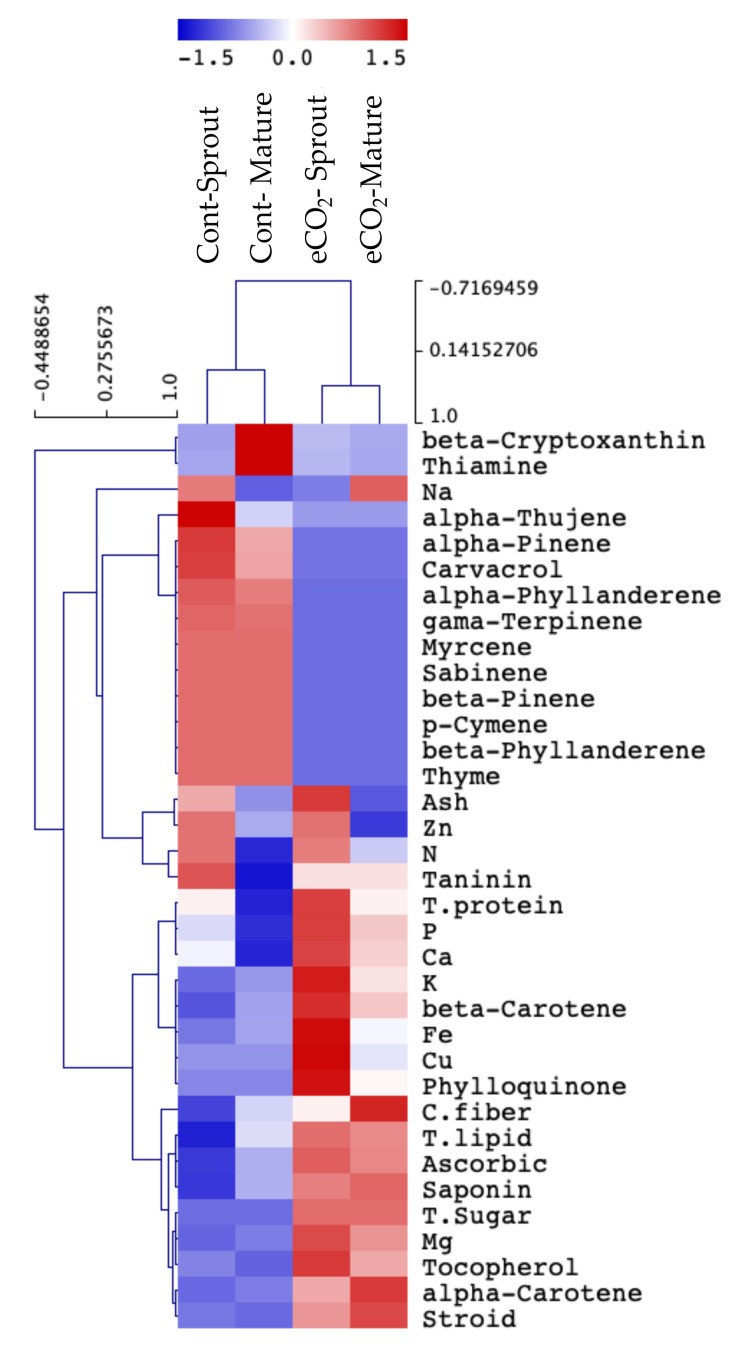
Hierarchical clustering analysis of total nutrients, minerals, vitamins and essential oils of control and eCO_2_-treated caraway sprouts and mature plants. Data are represented by the means of at least 3 replicates.

**Figure 3 plants-10-02434-f003:**
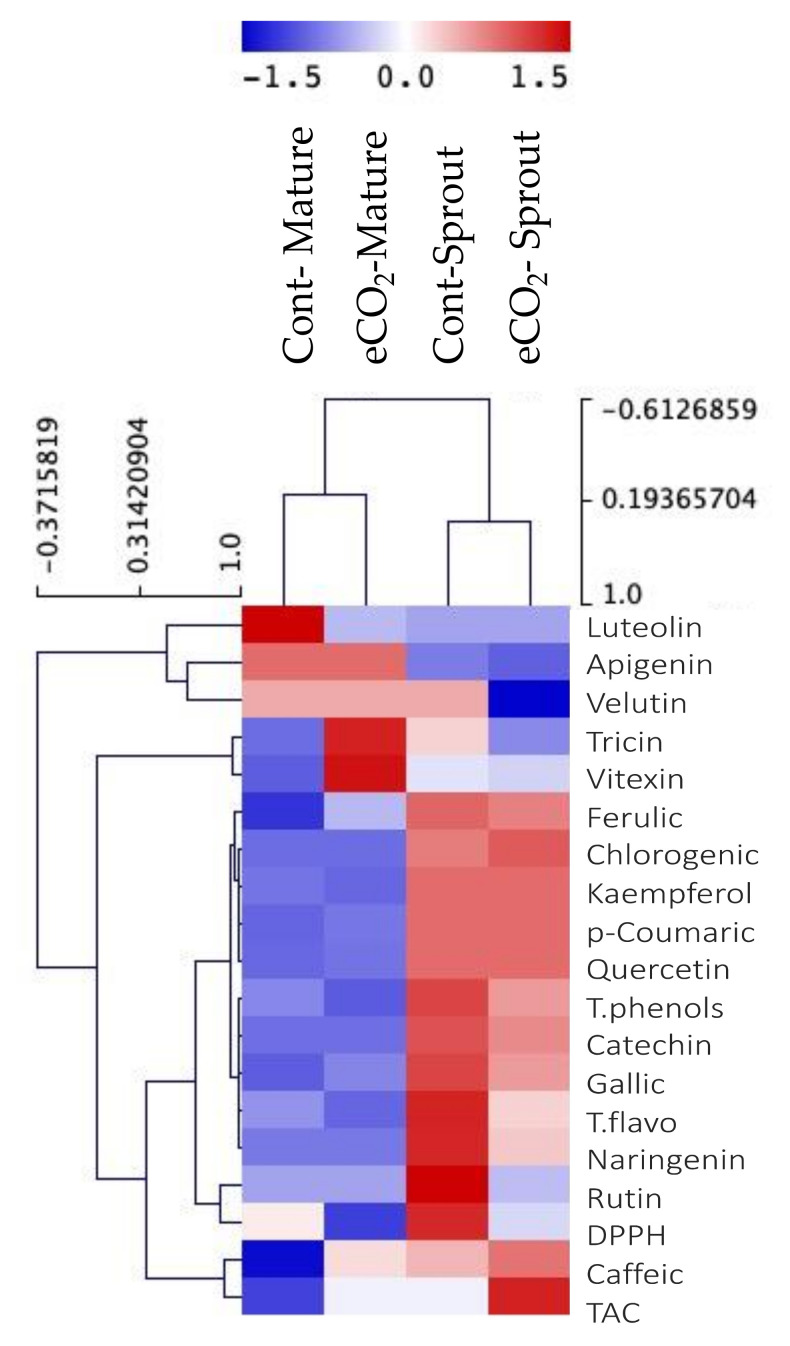
Hierarchical clustering analysis of phenolic compounds and antioxidant capacity of control and eCO_2_-treated caraway sprouts and mature plants. Data are represented by the means of at least 3 replicates.

**Figure 4 plants-10-02434-f004:**
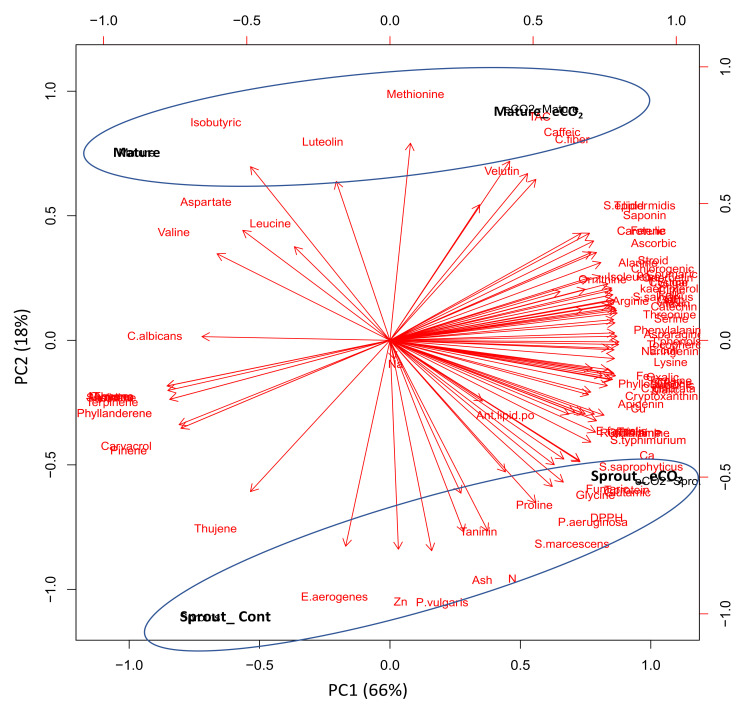
Principal component analysis (PCA) of chemical compositions and biological activities of caraway plants at two developmental stages (sprout and mature tissues) under control or eCO_2_ growth conditions.

**Table 1 plants-10-02434-t001:** Organic acids and amino acids of caraway plants at two developmental stages (sprouts and mature tissues) under control or eCO_2_ growth conditions. Data are represented by the means of at least 3 replicates ± standard deviations. Different small letter (a, b, c, and d) within a row indicate significant differences between control and eCO_2_-samples. One-way analysis of variance (ANOVA) was performed. Tukey’s test was used as the post-hoc test for the separation of means (*p* < 0.05).

Organic Acids	Sprouts		Mature	
(mg/g FW)	Control	eCO_2_	Control	eCO_2_
Oxalic	1.42 ± 0.14 a	1.99 ± 0.2 b	1.2 ± 0.10 a	1.91 ± 0.2 b
Malic	5.06 ± 0.31 a	6.49 ± 0.5 b	4.1 ± 0.7 a	6.09 ± 0.2 b
Succinic	2.84 ± 0.16 a	4.95 ± 0.43 b	2.3 ± 0.2 a	3.95 ± 0.41 b
Citric	2.45 ± 0.1 a	3.2 ± 0.39 b	2.5 ± 0.4 a	3.1 ± 0.19 b
isobutyric	1.06 ± 0.11 a	1.02 ± 0.01 a	1.16 ± 0.1 a	1.11 ± 0.01 a
Fumaric	0.42 ± 0.03 b	0.45 ± 0.01 b	0.32 ± 0.0 a	0.42 ± 0.0 b
Amino Acids (µg/g FW)				
Lysine	1.57 ± 0.13 a	2.62 ± 0.12 b	1.51 ± 0.11 a	2.12 ± 0.2 b
Histidine	1.61 ± 0.089 a	1.13 ± 0.01 a	1.61 ± 0.09 a	1.13 ± 0.01 a
Alanine	0.54 ± 0.04 a	1.23 ± 0.08 b	0.34 ± 0.041 a	1.7 ± 0.08 b
Arginine	0.98 ± 0.01 a	1.19 ± 0.09 a	0.78 ± 0.03 a	1.39 ± 0.1 b
Isoleucine	0.08 ± 0.00 a	0.13 ± 0.01 a	0.1 ± 0.00 a	0.11 ± 0.01 a
Asparagine	0.52 ± 0.04 a	0.91 ± 0.07 b	0.52 ± 0.05 a	0.77 ± 0.01 a
Ornithine	0.1 ± 0.02 a	0.12 ± 0.02 a	0.11 ± 0.03 a	0.11 ± 0.02 a
Glycine	0.6 ± 0.04 b	0.68 ± 0.06 b	0.2 ± 0.046 a	0.58 ± 0.01 b
Phenylalanine	0.16 ± 0.013 a	0.32 ± 0.01 b	0.11 ± 0.015 a	0.31 ± 0.04 b
Serine	0.18 ± 0.01 a	0.3 ± 0.02 b	0.19 ± 0.01 a	0.26 ± 0.01 b
Proline	0.58 ± 0.04 a	0.57 ± 0.05 a	0.52 ± 0.05 a	0.57 ± 0.05 a
Valine	0.24 ± 0.01 a	0.22 ± 0.03 a	0.34 ± 0.05 b	0.22 ± 0.03 a
Aspartate	0.02 ± 0.001 a	0.02 ± 0 a	0.025 ± 0.0 a	0.02 ± 0.0 a
Cystine	0.01 ± 0.0 a	0.1 ± 0.01 a	0.01 ± 0.0 a	0.1 ± 0.0 a
Leucine	0.16 ± 0 b	0.11 ± 0 a	0.13 ± 0.019 a	0.21 ± 0 b
Methionine	0.01 ± 0.001 a	0.01 ± 0 a	0.02 ± 0.002 b	0.031 ± 0 b
Threonine	0.05 ± 0.0 a	0.08 ± 0.01 b	0.043 ± 0.0 a	0.08 ± 0.01 b
Tyrosine	0.5 ± 0.03 a	0.66 ± 0.05 b	0.42 ± 0.039 a	0.61 ± 0.05 b
Glutamine	46.3 ± 3.5 b	52.79 ± 0.6 c	33.7 ± 4.2 a	51.1 ± 4.2 d
Glutamic acid	33.8 ± 2.5 b	46.2 ± 5.5 c	22.8 ± 5.8 a	33.2 ± 5.1 b

**Table 2 plants-10-02434-t002:** Antimicrobial activity of control and eCO_2_-treated caraway sprouts and mature plants. The activity is presented as the diameter of inhibition zone (mm). Data are represented by the means of at least 3 replicates ± standard deviations. Different small letter (a, b, c) within a row indicate significant differences between control and eCO_2_-samples. One-way analysis of variance (ANOVA) was performed. Tukey’s test was used as the post-hoc test for the separation of means (*p* < 0.05).

	Sprouts		Mature	
Microbial Name	Control (mm)	eCO_2_ (mm)	Control (mm)	eCO_2_ (mm)
*Staphylococcus saprophyticus*	15.34 ± 2.2 b	22.48 ± 1.8 c	11.14 ± 1.3 a	16.08 ± 1.29 b
*Staphylococcus epidermidis*	9.27 ± 1 a	20.13 ± 1.6 b	11.27 ± 1.2 a	22.1 ± 1.6 b
*Enterococcus faecalis*	14.57 ± 3.4 b	17.9 ± 0.5 c	9.52 ± 3.42 a	16.3 ± 0.56 b
*Streptococcus salivarius*	7.87 ± 1.15 a	16.14 ± 2.7 b	8.82 ± 1.12 a	13.11 ± 0.1 b
*Escherichia coli*	6.3 ± 0.9 b	9.68 ± 7.8 c	4.3 ± 0.9 a	9.68 ± 30.2 c
*Salmonella typhimurium*	11.17 ± 1.9 b	19.91 ± 3.1 c	7.17 ± 1.2 a	12.1 ± 1.1 b
*Pseudomonas aeruginosa*	20.02 ± 1.7 b	27.15 ± 1.5 c	14.02 ± 1.7 a	17.15 ± 1.1 b
*Proteus vulgaris*	20.8 ± 1.0 c	18.53 ± 1.82 b	10.8 ± 1.2 a	13.2 ± 1.5 ab
*Enterobacter aerogenes*	14.92 ± 0.3 c	13.96 ± 0.8 b	12.12 ± 0.9 ab	9.96 ± 0.17 a
*Serratia marcescens*	5.92 ± 0.2 b	7.68 ± 1.1 c	3.91 ± 0.96 a	4.68 ± 1.6 ab
*Aspergillus flavus*	13.93 ± 1.1 b	14.85 ± 1.8 b	7.91 ± 1.5 a	8.85 ± 1.0 a
*Candida albicans*	15.58 ± 5.07 b	15.76 ± 1.2 a	14.51 ± 1.01 a	14.2 ± 0.3 a
*Candida glabrata*	4.11 ± 0.954 a	8.11 ± 1.4 c	3.41 ± 0.92 a	5.91 ± 0.04 b

## Data Availability

Data presented in this study are available on reasonable request.

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
