# Peer review of "Effect of Elevated CO2 on Biomolecules’ Accumulation in Caraway (Carum carvi L.) Plants at Different Developmental Stages"

_plants, 2021, doi:10.3390/plants10112434_

Round 1
Reviewer 1 Report
This manuscript presents the results of an interestingly designed experiment to investigate changes in growth, photosynthesis and bioactive metabolites of Carum carvi at two developmental stages under control and elevated CO2 conditions. In addition, the authors in this study evaluated the effects of eCO2-induced changes in plant metabolites on the antioxidant and antimicrobial activities of caraway sprouts and mature plants.
I find this manuscript valuable and interesting, due to the fact that the results of the experiment may be useful to improve the nutritional value, functionality and health-promoting perspectives of caraway sprouts and mature plants. Nevertheless, I have few comments on it.
Keywords should be different from the words in the title.
Table 2: Salmonella typhimurium is included twice with different data, while Aspergillus flavus is missing. Which data in the table for this species is true? The same situation is in the text of the manuscript chapter 2.5., line 157-158, where Salmonella typhimurium is given twice. Which information for this species is correct, the one in lines 154-159 or in lines 165 and 166?
Line 163-165: "However, eCO2 significantly reduced the antimicrobial activity of caraway sprouts and mature plants against Proteus vulgaris, Enterobacter aerogenes, and Candida albicans, compared to normally cultivated controls." Did eCO2 really significantly reduce the antimicrobial activity of mature caraway plants against Proteus vulgaris and Candida albicans?
Figure 2: Improve the quality of the figure.
Figure 3: Please check if the trial order is correct, or it should not be the same as in Figure 2, i.e. Cont-Sprout, Cont-Mature, eCO2-Sprout, eCO2-Mature.
Figure 3: Galic or Gallic?
Line 394: "...according to AbdElgawad et al., (2019)" - this article should be included in References, and in the manuscript text, the item number from References should be given in square brackets instead of "(2019)".
Author Response
Dear Editor,
Thank you for your comments on our article manuscript entitled "Effect of elevated CO2 on biomolecules accumulation in caraway (Carum carvi L.) plants at different developmental stages". We have carefully read all comments and suggestions and made as much modifications as we believed to meet the reviewers’ requests. A detailed point by point response to reviewers’ comments is addressed at the end of this letter (typed in red).
Reviewer 1
This manuscript presents the results of an interestingly designed experiment to investigate changes in growth, photosynthesis and bioactive metabolites of Carum carvi at two developmental stages under control and elevated CO2 conditions. In addition, the authors in this study evaluated the effects of eCO2-induced changes in plant metabolites on the antioxidant and antimicrobial activities of caraway sprouts and mature plants.
I find this manuscript valuable and interesting, due to the fact that the results of the experiment may be useful to improve the nutritional value, functionality and health-promoting perspectives of caraway sprouts and mature plants. Nevertheless, I have few comments on it.
Thanks for positive feedback
Keywords should be different from the words in the title.
Response: Thanks for valuable comments; we have changed the keywords that were similar to those in the title
Table 2: Salmonella typhimurium is included twice with different data, while Aspergillus flavus is missing. Which data in the table for this species is true? The same situation is in the text of the manuscript chapter 2.5., line 157-158, where Salmonella typhimurium is given twice. Which information for this species is correct, the one in lines 154-159 or in lines 165 and 166?
Response: Thanks for valuable comments; the data of the data of the second Salmonella typhimurium is for Aspergillus flavus. Results in lines 154-159 or in lines 165 and 166 were checked and corrected.
Line 163-165: "However, eCO2 significantly reduced the antimicrobial activity of caraway sprouts and mature plants against Proteus vulgaris, Enterobacter aerogenes, and Candida albicans, compared to normally cultivated controls." Did eCO2 really significantly reduce the antimicrobial activity of mature caraway plants against Proteus vulgaris and Candida albicans?
Response: Thanks, we corrected this in the manuscript
Figure 2: Improve the quality of the figure.
Response: Thanks, done
Figure 3: Please check if the trial order is correct, or it should not be the same as in Figure 2, i.e. Cont-Sprout, Cont-Mature, eCO2-Sprout, eCO2-Mature.
Response: Thanks, the order is correct. We cluster both samples and measurements; thus the order of the samples was according to the cluster and we cannot change it.
Figure 3: Galic or Gallic?
Response: Thanks, corrected (Gallic)
Line 394: "...according to AbdElgawad et al., (2019)" - this article should be included in References, and in the manuscript text, the item number from References should be given in square brackets instead of "(2019)"
Response: Thanks, done

Reviewer 2 Report
Abstract
Maybe it will be advisable to tell the CO2 concentrations used in the control and eCO2, respectively. Providing this information in the Methods chapter is to late.
Lines 69-70:
„eCO2 was demonstrated to enhance photosynthesis, which ultimately encourages the biosynthesis of sugars in dark respiration process [20,22]. „ - This statement disagrees with current understanding in biochemistry: Respiration is the oxidative degradation of organic matter. It describes a membrane bound sequence of redox reactions aiming at energy production.
Lines 77, 100, etc
Please define the term „bioactive“ or delete it from your manuscript. (In this context, the term does not have any meaning.)
Fig. 1
Please provide information on units in use.
Line 105 & 108
In this chapter you are discussing CO2 effects on „nutrients“, such as sugars, proteins, etc. Cations, such as Ca, Mg, etc. you have named „minerals“. - In this context it is confusing, when you are calling trace cations, such as Cu, Fe, etc. „micro-nutrients“. Moreover, you are ending the chapter with calling K and P „macro-nutrients“. - Please decide on a clear terminology. Do you aim at feeding men or plants? - I suggest using your terminology from line 394.
Fig. 2
Please improve printing quality of this figure
chapter starting with line 220
This is not a discussion but a list of observations without any further explanation.
Fig. 4
Please make sure that this figure is readable. Maybe you will have to use numbers instead of words. Numbers will have to be explained in the figure legend.
Line 383
Just for curiosity: You have tested different homogenizers. Did you find any differences in reproduction of results, yield of extracts, etc.?
Author Response
Dear Editor,
Thank you for your comments on our article manuscript entitled "Effect of elevated CO2 on biomolecules accumulation in caraway (Carum carvi L.) plants at different developmental stages". We have carefully read all comments and suggestions and made as much modifications as we believed to meet the reviewers’ requests. A detailed point by point response to reviewers’ comments is addressed at the end of this letter (typed in red).
Reviewer 2
Abstract
Maybe it will be advisable to tell the CO2 concentrations used in the control and eCO2, respectively. Providing this information in the Methods chapter is to late.
Response: Thanks, we added this information to the abstract.
Lines 69-70:
eCO2 was demonstrated to enhance photosynthesis, which ultimately encourages the biosynthesis of sugars in dark respiration process [20,22]. „ - This statement disagrees with current understanding in biochemistry: Respiration is the oxidative degradation of organic matter. It describes a membrane bound sequence of redox reactions aiming at energy production.
Response: Thanks, we corrected it
Lines 77, 100, etc
Please define the term „bioactive“ or delete it from your manuscript. (In this context, the term does not have any meaning.)
Response: Thanks, we replaced it by other words in the whole manuscript.
Fig. 1
Please provide information on units in use.
Response: Thanks, we added the units to the figure legend
Line 105 & 108
In this chapter you are discussing CO2 effects on „nutrients“, such as sugars, proteins, etc. Cations, such as Ca, Mg, etc. you have named „minerals“. - In this context it is confusing, when you are calling trace cations, such as Cu, Fe, etc. „micro-nutrients“. Moreover, you are ending the chapter with calling K and P „macro-nutrients“. - Please decide on a clear terminology. Do you aim at feeding men or plants? - I suggest using your terminology from line 394.
Response: Thanks, we corrected them and used the terminology from line 394
Fig. 2
Please improve printing quality of this figure
Response: Thanks, done
chapter starting with line 220
This is not a discussion but a list of observations without any further explanation.
Response: Thanks; the discussion of these observations was already provided starting from line 229
Fig. 4
Please make sure that this figure is readable. Maybe you will have to use numbers instead of words. Numbers will have to be explained in the figure legend.
Response: Thanks; but using numbers will be confusing, so we have maximized the figure to be more readable
Line 383
Just for curiosity: You have tested different homogenizers. Did you find any differences in reproduction of results, yield of extracts, etc.?
Response: No, we did not find any differences, they were almost similar
